# A Fine-Grained Taxonomy of Replies to Hate Speech

**Xinchen Yu**[1]    **Ashley Zhao**[2]    **Eduardo Blanco**[1]    **Lingzi Hong**[3]

[1]University of Arizona
[2]TAMS, University of North Texas
[3]University of North Texas
xinchenyu@arizona.edu

## Abstract

Countering rather than censoring hate speech has emerged as a promising strategy to address hatred. There are many types of counterspeech in user-generated content: addressing the hateful content or its author, generic requests, well-reasoned counter arguments, insults, etc. The effectiveness of counterspeech, which we define as subsequent incivility, depends on these types. In this paper, we present a theoretically grounded taxonomy of replies to hate speech and a new corpus. We work with real, user-generated hate speech and all the replies it elicits rather than replies generated by a third party. Our analyses provide insights into the content real users reply with as well as which replies are empirically most effective. We also experiment with models to characterize the replies to hate speech, thereby opening the door to estimating whether a reply to hate speech will result in further incivility.

## 1 Introduction

Counterspeech refers to a "direct response that counters hate speech" (Mathew et al., 2019). It is a remedy to address hate speech (Richards and Calvert, 2000) by spelling out the hateful content conveyed in an utterance, or challenging, questioning, rejecting, disputing or confronting the hate (Langton, 2018; Goffredo et al., 2022). Distinct from approaches to hate mitigation by content moderation (Schmidt and Wiegand, 2017; Fortuna and Nunes, 2018), counterspeech is preferable as it does not interfere with the principle of free and open public spaces for debate (Schieb and Preuss, 2016; Chung et al., 2019).

Recently, the NLP communities have contributed corpora for the detection (Mathew et al., 2019; He et al., 2021; Albanyan and Blanco, 2022; Yu et al., 2022) and categorization (Mathew et al., 2019; Goffredo et al., 2022) of counterspeech. Categorizing

Hateful post: *You are full of fat f\*\*k liberals who jump to say a person died because they didn't wear a flimsy piece of cloth and not because they made awful life decisions and stuffed their faces.*

- Reply addressing the author: *You seem like don't like it here. So why are you loser here?* [subsequent hate: 2]

- Reply addressing the content: *The f\*\*king virus killed them, not the obesity. If they didn't get the virus, they would still be alive.* [subsequent hate: 0]

Table 1: An excerpt from two Reddit conversations. The first reply disagrees with the hateful post by attacking the author, and there are two additional hateful comments in the subsequent conversation. The second reply disagrees with the hateful post by addressing the content, and there are no subsequent hateful comments.

counterspeech into finer-grained categories has enhanced our understanding of interactions between hate speech and counterspeech. Therefore, it has the potential to identify effective interventions to deal with hate speech and further promote constructive discourses. Existing studies mostly focus on the identification or categorization of counterspeech and ignore replies that do not counter but may be effective in de-escalating online hatred (i.e., neutral replies). In addition, few studies working on categorizing counterspeech have (a) differentiated the target of counterspeech (i.e., addressing the author or the hateful content), or (b) grounded the categorization on argument mining theories.

Counterspeech addressing the author of hate speech can lead to completely different conversational outcomes than counterspeech addressing hateful content. Table 1 shows two replies countering a hateful post. While both replies use hostile language (Mathew et al., 2019), the former disagrees with the hateful post by denigrating the author ([. . . ] *you loser*), and results in two additional hateful comments in the subsequent conversation. The latter reply contradicts the content of the hateful post and backs it up with a credible rationale

| Authors | Source (Lang.) | Size | Balance | Hate | Categorization | Target |
|---|---|---|---|---|---|---|
| He et al. (2021) | Twitter (en) | 2,290 | 22.6% | | | |
| Vidgen et al. (2021) | Reddit (en) | 27,494 | 0.8% | ✓ | | |
| Albanyan and Blanco (2022) | Twitter (en) | 11,304 | 20.0% | ✓ | | ✓ |
| Yu et al. (2022) | Reddit (en) | 6,846 | 23.7% | ✓ | | |
| Mathew et al. (2019) | YouTube (en) | 11,093 | 49.5% | | ✓ | |
| Goffredo et al. (2022) | Reddit (it) | 624 | 13.0% | ✓ | ✓ | |
| Ours | Reddit (en) | 3,654 | 41.2% | ✓ | ✓ | ✓ |

Table 2: Summary of counterspeech datasets from user-generated web content in terms of Size, counterspeech class Balance ratio, and the inclusion of Hate comment, fine-grained counterspeech Categorization, and Target of counterspeech. We are the first to include a fine-grained categorization of counterspeech that spells out the target.

that does not result in additional hateful comments.

Addressing the author rather the content is usually considered a weaker form of disagreement (Graham, 2008). Previous works in argument mining have looked at counterarguments that attack either the author of an argument (Habernal et al., 2018) or the content of an argument (Wachsmuth et al., 2018; Jo et al., 2020; Alshomary et al., 2021). Counterarguments and counterspeech share a similar purpose: to present an alternative stance to a statement (Chung, 2022), but counterspeech is not necessarily a counterargument (see Section 2).

In this paper, we present a categorization of counterspeech that is well-grounded on theories in argument mining and builds a bridge between these two related yet unconnected research areas. Specifically, our categorization differentiates between counterspeech that addresses the author of hate speech and the hateful content itself. Therefore, it could provide useful insights into counterspeech generation and evaluation strategies. Importantly, our categorization can help uncover how language usage in different types of replies is tied to the future conversation trajectory.

The main contributions are as follows:[1]

- A theoretically-grounded taxonomy of replies to hate speech with categories indicating (a) whether a reply disagrees with hate speech and (b) whether replies that disagree address the author of hate speech or the content as well as secondary categories;
- Using the taxonomy to create an annotated dataset of replies to hate speech from Reddit;
- Comparing different types of replies to hate speech with respect to (a) language usage and (b) their relation to conversational outcomes;
- Building models to predict the primary and secondary categories a reply to hate speech

belongs to. Additionally, we present a qualitative error analysis.

## 2 Related Work

**Hate Speech** There have been several studies on hate speech, including efforts on defining taxonomies for hate speech. Examples include hate speech against both individuals and groups (Zampieri et al., 2019), against some specific identities (Waseem et al., 2017; Fortuna and Nunes, 2018), against only a single identity (Guest et al., 2021; Vidgen and Yasseri, 2020), or categories for implicit hate (ElSherief et al., 2021).

**Counterspeech** is a strategy to address hate speech that does not require content removal, and it has received increasing attention. Although a few works have built fine-grained counterspeech taxonomies (Mathew et al., 2019; Goffredo et al., 2022; Allaway et al., 2023), they do not differentiate between the target of the counterspeech (i.e., the author of hate or the hate itself). In addition, they fail to convey the intensity of hostile tone in counterspeech (Benesch et al., 2016; Mathew et al., 2019). For example, insulting somebody's intelligence is more uncivil than asking them to leave the discussion. There are a few synthetic datasets that have been curated with the help of trained operators (Qian et al., 2019; Chung et al., 2019; Fanton et al., 2021) or by generative models (Allaway et al., 2023). However, synthetic counterspeech is not as rich as genuine counterspeech and rarely used by real users. For example, "*[. . . ] are inappropriate*" and "*[. . . ] should be avoided*" made up 5.6% of replies in the synthetic counterspeech dataset by Qian et al. (2019). On the other hand, we found no real users using this kind of generic language to counter hate speech in our dataset. In this paper, we work with counterspeech written by regular people out of their own motivations and desires.

---

[1]Data available at https://github.com/xinchenyu/counter_taxonomy.

Table 2 summarizes existing counterspeech datasets from user-generated web content. We are the first to propose a fine-grained taxonomy of replies to hate speech that differentiates between (a) replies countering and not countering hate speech and (b) replies that counter hate speech by addressing the author of hate speech and the hateful content. Our work also complements recent efforts to capture and understand counterspeech replies that attack the author of the hate speech (Albanyan and Blanco, 2022). We note two differences. First, attacking the author is only a subset of addressing the author; we consider a broader set of scenarios, for example, accusing or blaming the author's behavior (namely, Accusation). Second, they consider two types of counterspeech, while we work with a fine-grained categorization.

**Counterspeech vs Counterargument** A counterargument is considered an ad-hominem argument if it argues against the author rather than the content (Habernal et al., 2018). Otherwise, it argues against the content of the argument by denying the premises, conclusion, or the reasoning between them (Walton, 2009; Wachsmuth et al., 2018). There are two main differences between counterspeech and counterarguments. First, counterspeech does not necessarily attack a hateful comment—it may provide suggestions to the author of the hateful comment (e.g., "*Please be nice to others*"). On the other hand, a counterargument always attacks either an argument or its author. Second, when addressing the content, a counterspeech comment does not require supporting evidence, while a counterargument usually does (Chung, 2022).

## 3 A Taxonomy of Replies to Hate Speech

We present a new taxonomy to categorize replies to hate speech that comprises four primary categories and eight secondary categories. It bridges the gap between argumentation theories (Habernal et al., 2018; Habernal and Gurevych, 2017; Wachsmuth et al., 2018) and counterspeech encountered in genuine user-generated web data. The taxonomy covers two dimensions of arguments: logos (i.e., logical arguments) and pathos (i.e., appealing to emotion) (Aristotle and Kennedy, 1991). Therefore, our categories are not mutually exclusive but represent principal types of replies to hate speech. By detailing argument components, this taxonomy facilitates (a) the evaluation of logical fallacies and emotional appeals in counterspeech

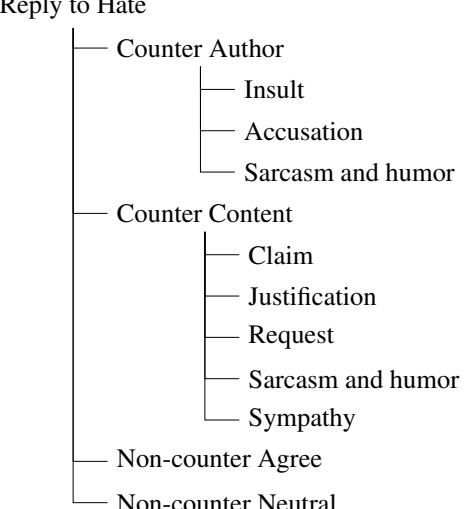

Figure 1: Taxonomy of replies to hate speech.

and (b) the analysis of how they relate to ineffective and uncivil communication. Figure 1 shows the taxonomic structure.

### 3.1 Counter Author

*Counter Authors* refers to the language that disagrees with the hateful comment by addressing the author or some features of the author's character (Tindale, 2007). The secondary categories comprise three sub-types: *Insult*, *Accusation*, and *Sarcasm and humor*.

**Insult** refers to language that explicitly attacks the author of the hateful comment, including (a) vulgar insults (Holgate et al., 2018; Habernal et al., 2018): "*your a\*\*hole*"; (b) intelligence insults (Habernal et al., 2018; Vidgen et al., 2021): "*so stupid I can't help you*"; (c) dehumanization (Leader Maynard and Benesch, 2016; Vidgen et al., 2021): "*you are just trash*"; or (d) threatening language (Zampieri et al., 2019): "*beat me if you dare*". Using insults is usually considered the worst way to disagree with hate speech (Graham, 2008).

**Accusation** refers to language that (a) blames or spells out the behavior or intention of the author of the hateful comment, including but not limited to lying (Holgate et al., 2018), trolling (Mihaylov and Nakov, 2016), ignorance (Jain et al., 2014), name-calling (Kenski et al., 2020), etc., (b) asks the author to leave the discussion, or (c) accuses the author of belonging to an affiliation or identity-based group (Vidgen et al., 2021) in a negative way (e.g., incels, radical right or left).

**Sarcasm and humor** refers to language that uses sarcasm (Waseem and Hovy, 2016; Justo et al.,

2014) or humor (Fortuna and Nunes, 2018; Goffredo et al., 2022) to address the author of the hateful comment. For example, "*Keep trying! You will make a good comment eventually*" uses sarcasm to mean the opposite of what it (explicitly) states.

## 3.2 Counter Content

*Counter Content* includes language that disagrees by addressing the content of the hateful comment. There are five sub-types in the secondary categories: *Claim*, *Justification*, *Request*, *Sarcasm and humor*, and *Sympathy*. We provide details below for these sub-types except for *Sarcasm and humor*, as the definition is similar as the definition of the sub-type by the same name under *Counter Author*. Indeed, the only difference is that the sarcasm or humor is directed towards the hateful comment.

**Claim** is an assertion put forward (Toulmin et al., 1979) that shows disagreement with the hateful comment by rebutting its premise, evidence, or conclusion, or targeting the reasoning between them (Walton, 2009) without providing any justifications or reasons for supporting the claim.

**Justification** refers to language that provides one or more justifications as evidence or reason to oppose the hateful content (Albanyan and Blanco, 2022). For example, the following reply is justified: "*This is racist. Non-white men are also represented in this infographic, not only the white men*".

**Request** is the language that (a) questions the validity of the content in the hateful comment and asks for more evidence or a justification (Walton, 2005; Chung et al., 2021), or (b) makes suggestions to the author of the hateful comment in a non-negative way (e.g. "*Hey there are definitely better ways to say it*"). We include *Request* in *Counter Content* based on our observations that making suggestions is usually closely related to the content of the hateful comment. Prior work includes speech that uses sympathy or kindly makes suggestions in the same category (namely, *positive tone*) (Mathew et al., 2019). Instead, we include the latter in *Request* and the former in a new sub-type, *Sympathy*.

**Sympathy** refers to language that expresses the feeling of sorrow or puts in a good word for someone being attacked in the hateful comment (Sosea and Caragea, 2022). The target being attacked by hate speech is sometimes the group the author of counterspeech affiliates with. For example, they may share identities or belong to the same affiliation-based group (Mathew et al., 2019), as in

"*It is so frustrating when people don't listen to him but try to paint a picture of him*".

## 3.3 Non-counter Agree / Non-counter Neutral

*Non-counter Agree* refers to agreeing with the hateful comment, including instances that not only agree but also include additional hate. On the other hand, *Non-counter Neutral* is language that neither agrees nor disagrees with the hateful comment. Some examples include shifting discussion topics, pointing to external sources, and sharing stories.

## 4 Data Collection and Annotation

We choose Reddit as the source of data and use the PushShift API to retrieve whole conversation threads.[2] To mitigate topic and author biases that keyword sampling may introduce (Wiegand et al., 2019; Vidgen et al., 2021), we use community-based sampling. We select 35 subreddits that are thought to be hateful (Qian et al., 2019; Guest et al., 2021; Vidgen et al., 2021), including subreddits such as r/MensRights, r/Seduction, and r/PurplePillDebate. We refer the reader to Appendix A for the full list. There are a total of 1,382,596 comments from 5,325 submissions.

## 4.1 Identifying Hateful Comments

As the prevalence of online hate in the wild is very low (0.1% in English language social media (Vidgen et al., 2021)), we first build three models to identify candidate comments with uncivil content in the 1,382,596 comments using pre-trained models (Liu et al., 2019) with the corpora by Davidson et al. (2017), Qian et al. (2019), and Vidgen et al. (2021). We consider a comment as candidate comment if any of the three classifiers predicts it to be uncivil. There are a total of 5,469 candidate uncivil comments. Uncivil comments cover broader cases than hateful comments (Davidson et al., 2020). To ensure quality, we validate candidates via crowdsourcing. We choose Amazon Mechanical Turk (MTurk) as the crowdsourcing platform. Annotators are provided with a detailed definition and examples of hate speech (Ward, 1997; Davidson et al., 2017; Vidgen et al., 2021). Only crowdworkers that pass a short 10-question qualification test checking for understanding can keep working on our task (minimum correct answers: 90%). Each candidate comment is annotated by three annotators. The Krippendorff's $\alpha$ coefficient (Krippen-

---

[2] https://pushshift.io/api-parameters/

| Primary | Secondary | # | % |
|---|---|---|---|
| Counter Author | Insult | 252 | 50 |
| | Accusation | 168 | 33 |
| | Sarcasm & humor | 83 | 17 |
| | Total | 503 | 100 |
| Counter Content | Claim | 361 | 36 |
| | Justification | 269 | 26 |
| | Request | 227 | 23 |
| | Sarcasm & humor | 79 | 8 |
| | Sympathy | 66 | 7 |
| | Total | 1,002 | 100 |
| Non-counter Agree | Total | 791 | 100 |
| Non-counter Neutral | Total | 1,358 | 100 |

Table 3: Percentage of replies in each category. *Non-counter Agree* and *Non-counter Neutral* do not have secondary categories, so only the total is shown.

| | p-value | Bonf. |
|---|---|---|
| **Textual factors** | | |
| Total tokens | ↓↓↓ | |
| Second pronoun | ↑↑↑ | ✓ |
| Question mark | ↓↓↓ | ✓ |
| Negation cues | ↓↓↓ | ✓ |
| Name entity (norp) | ↓ | |
| **Sentiment and social role factors** | | |
| Disgust words | ↑↑↑ | ✓ |
| Negative words | ↑↑↑ | ✓ |
| Positive words | ↓↓↓ | ✓ |
| Polite words | ↓↓↓ | ✓ |
| Male words | ↓↓↓ | ✓ |
| Female words | ↓↓↓ | ✓ |

Table 4: Linguistic analysis comparing the replies in *Counter Author* and *Counter Content* comments. Number of arrows indicates the p-value (Mann-Whitney U test; one: p<0.05, two: p<0.01, and three: p<0.001). Arrow direction indicates whether higher values correlate with *Counter Author* (up) or *Counter Content* (down). A check mark (✓) indicates that the statistical test passes the Bonferroni correction.

dorff, 2011) among a total of 26 annotators is 0.65, which indicates substantial agreement (Artstein and Poesio, 2008). The labels are obtained using the majority vote. 1,065 out of the total 5,469 candidate comments were labeled by the crowdworkers as hateful comments.

## 4.2 Annotating Replies to Hateful Comments

Next, we collect all the direct replies to each hateful comment that are 50 tokens or fewer. We reserve longer replies, which are more likely to fall into multiple categories, for future work. We label the replies using our fine-grained taxonomy (Section 3). A reply that falls into more than one subcategory will be labeled with multiple categories.

Since fine-grained categories are too subtle for MTurk workers, we hire three research assistants to be our expert annotators. The annotators underwent 4 weeks of training and are either native or fluent English speakers. There are a total of four annotation phases. In each phase, we collect annotations for 1,000 replies (except the last phase). In the first phase, we walk them through 100 replies, resolve disagreements, and refine confusing label definitions. We then let them independently work on the next 150 replies. Pairwise Cohen's $k$ between annotators is above 0.70, which is considered substantial (Artstein and Poesio, 2008), so each annotator then labels an independent partition of the data with 250 replies. We repeat the above steps for the next three phases to ensure high annotation quality and keep the time spent on annotations reasonable. The Krippendorff's $\alpha$ coefficients for the 150 double-annotated replies in each phase are 0.78, 0.73, 0.68, and 0.71, which are considered

substantial agreement. The final corpus consists of 3,654 replies to hateful comments, each of which is assigned a category in our taxonomy. Only 3% of the replies are labeled with multiple categories and excluded in the following analyses.

We refer the reader to Appendix B for a data statement providing more details about our corpus.

## 5 Corpus Analysis

Table 3 presents the percentage of labels that belong to primary and secondary categories. For the primary categories, the most frequent label is *Non-counter Neutral*, accounting for 37.2% of the replies followed by *Counter Content* (27.3%), *Non-counter Agree* (21.6%), and *Counter Author* (13.8%). To our surprise, when disagreeing with the hateful comment by addressing the author, most replies use *Insult* (50.1%). When disagreeing by addressing the content, *Claim* is the most frequent secondary category (36.0%).

### 5.1 Linguistic Cues

Past work has found that lexicon-based features can differentiate between *Counter* replies from *Non-counter* replies to hateful comments (Mathew et al., 2019; Albanyan and Blanco, 2022). In our study, we explore whether linguistic and lexicon-based features can distinguish *Counter Author* from *Counter Content*. We analyze the linguistic characteristics of replies to hateful comments when

| | Counter vs Non-counter | Author vs Content | One sub-type vs All the other terminal nodes | | | | |
|---|---|---|---|---|---|---|---|
| | | | Insult | Sarcasm | Justification | Request | Sympathy |
| Number of uncivil comments | ↑↑↑ | | ↑ | | ↑↑ | ↑↑ | ↑↑↑ |
| Number of civil comments | ↓↓↓ | ↓ | | ↑ | ↓↓ | ↑ | ↑↑↑ |
| Number of total comments | ↑↑↑ | | | ↑ | ↑↑↑ | ↑↑ | ↑↑↑ |

Table 5: Analysis of conversational outcomes by comparing the replies that are (a) *Counter* vs *Non-counter*, (b) when *Counter*, *Counter Author* vs *Counter Content*, and (c) a selected sub-type and all other terminal nodes in the taxonomy (i.e., the eight subtypes, *Non-counter Agree*, and *Non-counter Neutral*). We refer to the two subtypes named *Sarcasm and humor* as *Sarcasm*. Arrow direction indicates whether higher values correlate with the former in each comparisons (up). Tests having at least one arrow have passed the Bonferroni correction.

replies are *Counter Author* or *Counter Content* to shed light on the differences between the language people use when disagreeing with hateful comments. All factors we consider are based on counts of (a) textual features or (b) words indicating sentiment and social roles. We check for negation cues (Fancellu et al., 2016), mentions of all named entities,[3] and sentiment and social role factors using the Sentiment Analysis and Cognition Engine (SEANCE) lexicon (Crossley et al., 2017). We run Mann-Whitney U test (Mann and Whitney, 1947) and report the results in Table 4.

We observe several interesting findings:

- Regarding textual features, disagreeing by addressing the author uses more second pronouns, while addressing the content uses more tokens, question marks, and negation cues.
- Regarding sentiment and social role features, there are significantly more negative and disgusting words in replies when addressing the author, as well as less positiveness, politeness, and words related to the social roles of men (i.e., *buddy*, *boy*, *actor*) and women (i.e., *aunt*, *bride*, *daughter*).

## 5.2 Conversational Outcomes

Do replies to hateful comments result in more or less civility in follow-up conversations according to our fine-grained taxonomic categorization? Similarly to the previous analysis, we look for statistically significant differences in conversational outcomes between (a) *Counter* and *Non-counter* groups, (b) *Counter Author* and *Counter Content* groups, and (c) selected sub-types replies and all other terminal nodes in the taxonomy (i.e., the eight subtypes, *Non-counter Agree*, and *Non-counter Neutral*). We consider several properties of the subsequent conversation after a reply: (a) the number of uncivil comments, (b) the number of civil comments, and (c) the number of total comments. Uncivil comments have been identified in Section 4.1; all the other comments are considered civil. We verified the reliability of this method by manually annotating a sample of 150 comments and obtained Cohen's $k = 0.67$ between the manual and automated labels. This is considered substantial agreement and suggests that the automated labeling is a sound measure of whether a comment is civil or uncivil. Results are shown in Table 5.

**Counter vs Non-counter** Replies that disagree with the hateful comments (*Counter*) elicit (a) fewer civil comments and (b) more uncivil comments in the subsequent conversations. Thus combating hatred may elicit wider discussions but at the same time introduce additional uncivil behaviors.

**Counter Author vs Counter Content** When disagreeing with hateful comments, addressing the author correlates with fewer civil comments. This indicates that arguing against hatred in public online conversations may escalate hatred, but addressing the hate speakers may be even worse.

**A selected sub-type vs all other terminal nodes** Regarding the number of uncivil comments, *Insult* replies correlate with more subsequent uncivil comments. Surprisingly, we found that some *Counter Content* sub-types may attract more follow-up discussions and incubate additional uncivil behaviors (*Justification*, *Request*, and *Sympathy*). Our findings are consistent with previous work showing that when correcting misstatements, language toxicity increases (Mosleh et al., 2021).

## 6 Experiments

We experiment with models to solve two tasks:

1. Primary category classification. Given a reply to hateful comment, these models determine

---

[3]https://spacy.io/usage/linguistic-features

| | Non-counter | | | | | | Counter Author | | | Counter Content | | | W. Average | | |
|---|---|---|---|---|---|---|---|---|---|---|---|---|---|---|---|
| | Agree | | | Neutral | | | | | | | | | | | |
| | P | R | F1 | P | R | F1 | P | R | F1 | P | R | F1 | P | R | F1 |
| Majority | 0.00 | 0.00 | 0.00 | 0.36 | 1.00 | 0.53 | 0.00 | 0.00 | 0.00 | 0.00 | 0.00 | 0.00 | 0.13 | 0.36 | 0.19 |
| Random | 0.24 | 0.24 | 0.24 | 0.36 | 0.26 | 0.30 | 0.14 | 0.23 | 0.18 | 0.20 | 0.21 | 0.21 | 0.26 | 0.23 | 0.24 |
| Hate | 0.40 | 0.23 | 0.29 | 0.46 | 0.68 | 0.55 | 0.50 | 0.09 | 0.15 | 0.38 | 0.47 | 0.42 | 0.43 | 0.43 | 0.39 |
| Reply† | 0.53 | 0.34 | 0.42 | 0.55 | 0.59 | 0.57 | 0.70 | 0.52 | 0.59 | 0.43 | 0.59 | 0.49 | 0.54 | 0.52 | 0.52 |
| + pretrain† | **0.46** | **0.55** | **0.50** | 0.54 | 0.58 | 0.56 | 0.71 | 0.55 | 0.62 | 0.54 | 0.48 | 0.51 | 0.55 | 0.54 | 0.54 |
| + blend† | 0.46 | 0.48 | 0.47 | 0.55 | 0.67 | 0.60 | 0.78 | 0.56 | 0.65 | 0.51 | 0.43 | 0.47 | 0.56 | 0.55 | 0.55 |
| Hate+Reply†‡ | **0.56** | **0.45** | **0.50** | 0.58 | 0.70 | 0.63 | **0.74** | **0.59** | **0.66** | 0.56 | 0.56 | 0.56 | 0.60 | 0.59 | 0.59 |
| + pretrain†‡ | **0.52** | **0.48** | **0.50** | 0.61 | 0.56 | 0.58 | 0.68 | 0.59 | 0.63 | 0.51 | 0.65 | 0.57 | 0.57 | 0.57 | 0.57 |
| + blend†‡ | 0.53 | 0.45 | 0.49 | **0.61** | **0.71** | **0.66** | **0.74** | **0.59** | **0.66** | 0.56 | 0.59 | 0.58 | **0.61** | **0.60** | **0.60** |

Table 6: Results obtained with several models. We indicate statistical significance (McNemar's test (McNemar, 1947) over the weighted average) as follows: † indicates statistically significant ($p < 0.05$) results with respect to the *Hate* model, and ‡ with respect to the *Reply* model. We only show results pretraining and blending with the best instances (stance and EDA using only the *Reply* and pretraining and blending respectively, and counterspeech and EDA using *Hate+Reply* and pretraining and blending respectively). Training with the *hate comment + reply* coupled with blending and data augmentation yields the best results (F1: 0.60).

if it is *Counter Author*, *Counter Content*, *Non-counter Agree*, or *Non-counter Neutral*;

2. Counterspeech sub-type classification. Given a reply that disagrees with a hateful comment (i.e., *Counter Author* or *Counter Content*), these models determine the sub-type of counterspeech from our ontology.

We use a 70-15-15 split for each task. All of our models are neural classifiers with the RoBERTa transformer (Liu et al., 2019) as the main component. We use the pretrained models by Hugging-Face (Wolf et al., 2020) and Pytorch (Paszke et al., 2019) to implement our models.

## 6.1 Primary Category Classification

The neural classifiers consist of the RoBERTa transformer and another two fully connected layers to make predictions. To find out whether adding the hate comment would be beneficial, we consider three textual inputs: (a) the hate comment; (b) the reply to the hate comment; and (c) the hate comment and the reply. When considering both the hate comment and the reply, we concatenate them using [SEP] special token. The baseline models we use are the majority and random baselines. For the former, the majority label is predicted (*Non-counter Neutral*, Table 5). For the latter, a random label out of the four primary categories is predicted.

In addition to standard supervised learning, we explore two strategies to improve performance:

**Pretraining with Related Tasks** We experiment with several corpora to investigate whether pretraining with related tasks is beneficial. Pretraining takes place prior to training with our task. The related corpora that we use are hate speech (Davidson et al., 2017), sentiment (Rosenthal et al., 2017), sarcasm (Ghosh et al., 2020), counterspeech (Yu et al., 2022), and stance (Pougué-Biyong et al., 2021).

**Blending Additional Data** We also experiment with a complementary approach: blending additional corpora during the training process (Shnarch et al., 2018). We use both our corpus and additional corpora to train for $m$ blending epochs, and then use only our corpus to train for another $n$ epochs. Besides re-using the corpora used for pretraining (see above), we also create additional instances by adapting EDA (Easy Data Augmentation) to augment our own corpus (Wei and Zou, 2019). Specifically, we use Synonym Replacement, Random Insertion, Random Swap, and Random Deletion.

### 6.1.1 Quantitative Results

Table 6 shows the results per label and weighted averages. We provide here results pretraining and blending with the most beneficial tasks: stance for pretraining and EDA for blending. Using only the reply as input offers competitive performance with F1 scores up to 0.52 compared with the random baseline (F1: 0.52 vs. 0.24). Using both the hate comment and the reply as input yields much better results (F1: 0.59 vs 0.52). Consistent with previous work (Yu et al., 2022), this indicates that including the conversational context does help. Finally, the network that blends the corpus augmented by EDA

| Error Type | % | Example | Ground Truth | Predicted |
|---|---|---|---|---|
| Rhetorical question | 26 | Hate: *F\*\*k worthless inbreds who've contributed nothing to society.*
Reply: *Where are your contributions? I doubt there's any.* | Author | Content |
| Irony | 21 | Hate: *Retarded republicans fear everything.*
Reply: *It's amazing how broken you have to be to believe in their positions as a whole.* | Author | Agree |
| Discourse information | 18 | Hate: *Nothing. These hoes are just glorified bums. Treat em like they act... low-grade prostitutes.*
Reply: *At least prostitutes are honest about it.* | Agree | Neutral |
| General knowledge | 8 | Hate: *This comment section is smelling of f\*\*king Incels. But this girl is a loser. Put her back in jail.*
Reply: *Last I checked, 'incels' were people who think they're unlovable, not guys who hate attempted murder.* | Content | Neutral |
| Metaphorical language | 8 | Hate: *This b\*\*ch built like a toad.*
Reply: *She reminds me of a Porsche 911.* | Agree | Neutral |

Table 7: Most common error types made by the best model (predictions by *hate comment + reply + blending*).

|  | P | R | F1 |
|---|---|---|---|
| Insult | 0.62 | 0.66 | 0.64 |
| Accusation | 0.50 | 0.28 | 0.36 |
| Sarcasm and humor | 0.30 | 0.33 | 0.32 |
| Claim | 0.61 | 0.55 | 0.58 |
| Justification | 0.64 | 0.65 | 0.65 |
| Request | 0.69 | 0.73 | 0.71 |
| Sympathy | 0.21 | 0.60 | 0.32 |
| Weighted Average | 0.58 | 0.57 | 0.57 |

Table 8: Results of counterspeech type classification obtained with the best model.

and takes both the hate comment and the reply as input yields the best results (F1: 0.60).

### 6.1.2 Qualitative Analysis

To further understand the challenges in primary category classification, we manually analyze 100 random errors made by our best model (Hate+Reply+blend). We discovered a set of error types grounded on the language used in the replies and hateful comment. Table 7 lists and exemplifies the most common error types.

Rhetorical questions (Schmidt and Wiegand, 2017) are the most common (26%) type of errors. In the example, the model fails to realize that the reply does not ask for further validation but instead accuses the author of the hateful comment making no contributions. Irony (Nobata et al., 2016; Qian et al., 2019) is also a common error type (21%) and requires reasoning and understanding. The example uses irony to attack the author instead of showing agreement ("*amazing how broken*").

We also found that 18% of errors occur when discourse information (de Gibert et al., 2018) is re-

quired to make correct predictions. In other words, the reply itself looks neutral and does not contain hate speech, but the combination of both the hateful comment and the reply does convey hatred. Errors may also occur when general knowledge is required to understand hateful content (8%) or when the reply uses Metaphorical language (ElSherief et al., 2021) to implicitly express hatred (5%). In the bottom example from Table 7, a woman is referred to as a "*toad*" and "*Porsche 911*".

### 6.2 Counterspeech Sub-type Classification

We further build models for a 7-way classification task in which the input is a *Counter Author* or *Counter Content* reply along with the hate comment and the output is the sub-type of counterspeech presents in the reply. Table 8 shows the results. We retrain the best-performing system from Table 6: (Hate + Reply + blending). The *Sarcasm and humor* class in *Counter Author* and *Counter Content* are combined together considering that the prevalence of both classes in our corpus is low (*Counter Author*: 2.3%; *Counter Content*: 2.2%). Results show that predicting fine-grained counter sub-type for a Counter reply is a challenging task (F1: 0.57). Regarding recall, the model performs better when predicting *Request*, *Justification*, or *Insult*, compared with *Accusation* and *Sarcasm and humor*.

### 7 Conclusion

In this paper, we introduce a theoretically-grounded taxonomy of fine-grained categories of replies to hate speech. The taxonomy allows us to differentiate replies that disagree by addressing the author

of hate speech from addressing the content. We annotate a not-large-but-substantial amount of Reddit comments based on this taxonomy, thereby enabling the research community to better understand and model replies to hateful comment.

We further analyze whether different types of replies to hate speech differ in linguistic features and conversational outcomes in the subsequent conversations. We find that disagreeing with the hatred by addressing the author uses less positive and polite words, which indeed relates to less civil conversational outcomes compared with disagreeing by addressing the content. Experimental results of a 4-way classification are not perfect but encouraging: taking the hateful comment into account coupled with blending and data augmentation yields significant improvements. Our qualitative analysis of the most common errors show that a lot of errors are made when replies contain rhetorical questions or use irony to show disagreements.

Our findings point towards several opportunities for promoting healthier interactions in online platforms. For example, while our results show that disagreeing with hate speech may elicit additional incivility, we never claim that counterspeech is counterproductive. Counterspeech effectiveness can be further assessed based on changes in behavior or beliefs undergone by authors of hate speech. User studies are needed to guide such research line. Our results have further revealed that replies that kindly make suggestions or show empathy towards the target of hate speech may attract more follow-up discussions and more uncivil comments. Additional work is needed to better understand this phenomenon and whether it is present across different types of hate speech, for example, misogyny, homophobia, or Islamophobia.

## Limitations

Our work has several limitations. One is that we identify uncivil comments automatically with classifiers. Although we validate the reliability of labels obtained by classifiers with a small sample, some uncivil comments that we work with are actually not uncivil. Likewise, we are unable to capture some uncivil comments, for example, implicit hate speech (ElSherief et al., 2021). Additionally, our corpus consists of 3,654 replies to hate speech, which is not large. The samples are limited as our manual annotation effort needs annotators to understand and label the fine-grained categories and

resolve disagreements, which requires expertise knowledge and is quite time-consuming. We focus on replies that fall into only one category, as most comments on social media are short. However, there are also replies that adopt a combination of strategies to respond to hatred, for example, providing justification and showing sympathy at the same time. We reserve such complicated cases for future studies. Finally, while we have set a qualification test and provide detailed instructions to crowdworkers and in-house annotators, people's perceptions of hate speech may still vary slightly. We obtain the ground truth of hateful comments using majority vote.

## Ethical Considerations

The study has been through careful consideration of the risks and benefits to ensure that the research is conducted in an ethical manner. First, there are no identifiable participants in the process. The data we collected from Reddit is public available and is further anonymized by removing user names and other personally identifiable information for storage, annotation, and analysis. Second, we have informed crowdworkers and research assistants about the annotation tasks and obtained their content to participate. They were warned in the instructions that the content might be offensive or upsetting. They were also encouraged to stop the annotation process whenever the felt upset or overwhelming. We compensated them with $8.5 per hour. Third, the examples in this work are included to showcase the severity of the problem with hate speech. They are taken from actual web data and in no way reflect the opinion of the authors. The data will be shared based on methodological, legal, and ethical considerations (Weller and Kinder-Kurlanda, 2016). Finally, we also acknowledge the risk associated with releasing the dataset. However, we believe the benefit of shedding light on how people react to hatred outweighs any risks associated with the dataset release.

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

## A  Subreddit List

We provide here the list of subreddits we have selected: r/antiwork, r/antheism, r/bakchodi, r/bindingofisaac, r/changemyview, r/conspiracy, r/DankMemes, r/Drama, r/FemaleDatingStrategy, r/Feminism, r/GenZedong, r/HermanCainAward, r/justneckbeardthings, r/NoFap, r/KotakuInAction, r/ImGoingToHellForThis, r/MensRights, r/Sino, r/MetaCanada, r/modernwarfare, r/Seduction, r/playrust, r/PurplePillDebate, r/PussyPass, r/PussyPassDenied, r/ShitPoliticsSays, r/4Chan, r/ShitRedditSays, r/worldnews, r/SubredditDrama, r/TrueReddit, r/DotA2, r/TumblrInAction, r/TwoXChromosomes, r/BlackPeopleTwitter.

## B  Data Statement

As per the recommendations by Bender and Friedman (2018), we provide a data statement to better understand the new data presented in this paper.

### B.1  Curation Rationale

We create a new dataset to interpret replies to hate speech in user-generated web content. The dataset includes hateful comments and their direct replies with annotations indicating the fine-grained category of the replies.

To identify hateful comments, we first select 35 subreddits considered to be hateful and scrape whole conversation threads from these subreddits. As the prevalence of online hate in the wild is very low, we further build pretrained language models to automatically identify candidate uncivil comments. A set of 5,469 candidate uncivil comments is then collected based on our previous efforts. Finally we assign all the candidate uncivil comments to annotators for validation. Inter-annotator agreement (Krippendorff's $\alpha$) is 0.65, indicating substantial agreement; over 0.8 would be nearly perfect. We collect 1,065 candidates comments that were labeled as hateful.

After identifying hateful comments, we collect direct replies to them and manually annotate their types in the taxonomy we work with. Three annotators participate in the annotations. All the 3,654 replies to hateful comments were annotated by at least one annotator. We set a total of four phases. In each phase, we randomly select 150 replies to be annotated by three annotators. If inter-annotator agreement is above 0.60, which is substantial agreement, then each annotator works on an independent partition of the replies. The final version of the rules used to scrape comments, identify hateful comments and annotate replies to hateful comments are detailed in Section 4.

### B.2  Language Variety

The data collection process was carried out from April to July 2022. In Reddit, more than 95% of comments are written in English. We also use SpaCy to make sure that each reply is in English. Information on the specific type of English is not available.

### B.3  Speaker Demographic

The 3,654 replies along with the hateful comments are posted by Reddit users. We do not require comments to come from verified accounts. As per Reddit age restrictions, the minimum user age is 13 years for both authors of hateful comments and replies. Speakers are not reachable and thus demographic information about the speakers is limited.

### B.4  Annotator Demographic

Three annotators are part of the annotation process and development of annotation guidelines. All of

them are women and their ages range from 18 to 35 years old. Ethnic backgrounds are as follows: 2 are from Asian and 1 is from North America. All of the annotators are highly proficient in English. Socioeconomic backgrounds are as follows: all annotators reported that they are in the middle class. Educational background are as follows: 1 undergraduate, 1 graduate student, and 1 with a doctoral degree. 2 of them work in NLP-related research areas, and 1 majors in computer science.

All crowd workers are self-reported to be over 18 years old. They need to pass Adult Content Qualification in order to work on our tasks. The other demographic information is limited.

### B.5 Speech Situation

Text in our corpus is retrieved from Reddit between April to July of 2022. Modality of text is written by users on Reddit. Reddit allows users to edit their comments, and we use the version of the comments available as of July 2022. The interactions are asynchronous and replies and hateful comments cannot appear in Reddit simultaneously. The intended audience could be any user on the internet.

## C  Training Details

Our dataset was pre-processed by removing URLs, removing symbols, removing any additional spaces, and at the end, converting all words to lower-case. The neural model takes about half an hour on average to train on a single NVIDIA TITAN Xp.

We use the implementation by Pruksachatkun et al. (2020) and fine-tune the RoBERTa (base architecture; 12 layers) (Liu et al., 2019) model for each of the three training settings. For each setting, we set the hyperparameters to be the same when the input is the hateful comment, the reply, or both (Table 9).

|           | Epochs | Batch size | Learning rate | Dropout | Patience |
|-----------|--------|------------|---------------|---------|----------|
| reply     | 5      | 8          | 1e-5          | 0.5     | 10       |
| + pretrain | 5     | 8          | 1e-5          | 0.5     | 10       |
| + blend   | 2      | 4          | 1e-5          | 0.5     | 10       |

Table 9: Hyperparameters used to fine-tune RoBERTa individually for each training setting. We accept default settings for the other hyperparameters as defined in the implementation by Pruksachatkun et al. (2020).