# OpenReview forum: "A Fine-Grained Taxonomy of Replies to Hate Speech"
_EMNLP/2023/Conference — EMNLP 2023 Main_

### Official Review · Reviewer_hSjz · 2023-08-03

**Typos Grammar Style And Presentation Improvements:** 1. For the statement in the third par…
**Soundness:** 4

**Excitement:**

3: Ambivalent: It has merits (e.g., it reports state-of-the-art results, the idea is nice), but there are key weaknesses (e.g., it describes incremental work), and it can significantly benefit from another round of revision. However, I won't object to accepting it if my co-reviewers champion it.

**Missing References:**

1. In line 125 ( "..." is generic and rarely used by real users, but common in synthetic counterspeech), references or statistics are needed.

**Paper Topic And Main Contributions:**

This paper proposes a fine-grained taxonomy of replies to hate speech that addresses hateful content or its author. This taxonomy is applied to a Reddit dataset the authors collected. It also provides corpora and linguistic analyses of the interaction between hateful content and its replies, giving insights into what replies can elicit civil/uncivil outcomes.

**Questions For The Authors:**

1. Does Non-counter Agree / Non-counter Neutral include supporting victims?

**Reasons To Accept:**

1. A taxonomy of replies to hate speech based on argumentation theories which can encourage studies on projecting persuasion to the design of counterspeech that can be potentially effective in hate mitigation. However, I think this taxonomy may be more applicable to specific platforms like Reddit or forums compared to Twitter in which not much argumentative discussion is present in hate speech domain.
2. An analysis that looks into the dynamics of hate speech and replies

**Reasons To Reject:**

The paper lacks discussion around existing taxonomies for categorising counterspeech: how existing taxonomy compares with the one proposed, and why a new taxonomy is needed (is it really needed?). What types of benefits does this new taxonomy bring and what limitations exist in the existing taxonomies? For instance, several categories in the proposed taxonomy overlap with existing taxonomy like Benesh's [1], which are generally expressed in different/argumentative terms (e.g., claims vs facts/hypocrisy; accusation vs denouncing). For this reason, it is not clear why and how this new taxonomy can add knowledge to what we already knew about counterspeech.

[1] Benesch, S., Ruths, D., Dillon, K. P., Saleem, H. M., and Wright, L. (2016). Counterspeech on Twitter: A field study. Dangerous Speech Project.


**Reproducibility:**

4: Could mostly reproduce the results, but there may be some variation because of sample variance or minor variations in their interpretation of the protocol or method.

**Reviewer Confidence:**

5: Positive that my evaluation is correct. I read the paper very carefully and I am very familiar with related work.

---

> ### Author Rebuttal · Authors · 2023-08-27
>
> **Reasons to Reject**
>
> RE: The paper lacks discussion around existing taxonomies for categorising counterspeech: how existing taxonomy compares with the one proposed, and why a new taxonomy is needed (is it really needed?).
> * Regarding counterspeech, CONAN and the work by Mathew et al., (2019) mainly built on or simply followed the taxonomies by Benesch et al., (2016) with slight modifications. We conducted detailed analyses of previous taxonomies on counterspeech to ensure the new taxonomy covers all different types of counterspeech. There are overlaps in some categories (i.e., line 193, 208, 248, 258), however, the previous taxonomies fail to convey some important information. Most notably:
>     * the intensity of hostile tones does vary: intelligence insults [Insult] are more uncivil than asking someone to leave the discussion [Accusation].
>     * the strategies of positive tones can be different: kindly provide suggestions [Request] and show sympathy [Sympathy].
> * Our work has two salient distinctions: (1) we indicated whether the counterspeech addresses the author of hate speech or the content (i.e., in Table 2) and (2) we formalized the taxonomy based on argumentation theories (i.e., line 165-172). These yield several advantages:
>     * In counterspeech generation, Counter-content is generated by focusing on the content of hate speech, while Counter-author may have greater flexibility.
>     * In the evaluation, the relevance of the content in counterspeech to hate speech is a crucial factor for quality evaluation. But this factor may not be necessarily required for counterspeech.
>     * For implications, Counter-content might provide guidance on what and why the content is inappropriate. While Counter-author is usually not that informative.
> * Thank you for the excellent suggestion, we will revise accordingly to make the distinctions more straightforward and clearer (Table 2 and Section 2).
>
>
> **Questions for the Authors**
>
> RE: Does Non-counter Agree / Non-counter Neutral include supporting victims?
> * Supporting or saying good words for victims would belong to Sympathy (line 251-260).
>
>
> **Missing References**
>
> RE: In line 125 ( "..." is generic and rarely used by real users, but common in synthetic counterspeech), references or statistics are needed.
> * Thank you! We will add the following in Section 2: We observe “[...] are inappropriate” and “should be avoided” made up of 5.6% of the replies in the synthetic counterspeech dataset (Qian et al., 2019). While in the comments by real users, we found no one actually uses these kinds of language.
>
> **Typos Grammar Style And Presentation Improvements**
>
> RE: For the statement in the third paragraph of the introduction, I reckon this connection/assumption is a bit too strong and would require large-scale analysis or relevant evidence to back it up. [...] It may be better to include relevant references or soften the statement.
> * Thank you for the suggestion. We will tone down the statement.
>
> RE: The sentence in lines 355-357 can be deleted as it is repeated in lines 341-343.
> * Thank you for the suggestion. We will delete the content accordingly.
>
> RE: Line 542: in our corpus is low
> * Thank you. We will fix this typo and proofread our paper carefully.

---

### Official Review · Reviewer_LEi4 · 2023-08-05

**Soundness:** 4

**Excitement:**

4: Strong: This paper deepens the understanding of some phenomenon or lowers the barriers to an existing research direction.

**Paper Topic And Main Contributions:**

This paper studies “countering” as a form of replies to initial hatespeech content. “Counters” can be of various forms – counter arguments, insults, etc,.
This work categorizes the types of “counters” into a taxonomy, and further investigates the language usage, and if the final outcome of the conversation remains hateful.


**Reasons To Accept:**

1.	Well studied theoretical work to propose taxonomy for Counterspeech, being the first dataset to propose a fine-grained categorization of counterspeech including the target of counterspeech.
2.	Manual annotation efforts to create dataset of 3654 samples.
3.	This work studies overlooked conversational patterns in the form of counterspeech.


**Reasons To Reject:**

1. It would give the study depth if a larger (noisy) dataset were also created for experiments of weak supervision, to further investigate the value of the manually annotated data.

**Reproducibility:**

4: Could mostly reproduce the results, but there may be some variation because of sample variance or minor variations in their interpretation of the protocol or method.

**Reviewer Confidence:**

4: Quite sure. I tried to check the important points carefully. It's unlikely, though conceivable, that I missed something that should affect my ratings.

---

> ### Author Rebuttal · Authors · 2023-08-27
>
> **Reasons to Reject**
>
> RE: It would give the study depth if a larger (noisy) dataset were also created for experiments of weak supervision, to further investigate the value of the manually annotated data.
> * This is an excellent suggestion that we have thought of earlier. We would be interested to explore this method in our future work.

---

### Official Review · Reviewer_v5qd · 2023-08-11

**Soundness:** 3

**Excitement:**

3: Ambivalent: It has merits (e.g., it reports state-of-the-art results, the idea is nice), but there are key weaknesses (e.g., it describes incremental work), and it can significantly benefit from another round of revision. However, I won't object to accepting it if my co-reviewers champion it.

**Missing References:**

[1] Allaway, E., Taneja, N., Leslie, S.J. and Sap, M., 2023. Towards countering essentialism through social bias reasoning. arXiv preprint arXiv:2303.16173.
[2] Maarten Sap, Saadia Gabriel, Lianhui Qin, Dan Jurafsky, Noah A Smith, and Yejin Choi. 2020. Social bias frames: Reasoning about social and power implications of language.


**Paper Topic And Main Contributions:**

The authors introduce a taxonomy of counterspeech replies to hate speech with a classification on countering personal attacks from the author to countering the content of the hate speech reply or refraining from countering, grounded in social psychology. They annotate a decent sample of Reddit comments based on this taxonomy and analyze the types of replies used by humans in the wild.


**Questions For The Authors:**

1. With the Section 6 Experiments, it is clearly hard to model or classify these categories. How do you propose to improve on these models besides naive finetuning or augmentation with EDA? Do you see a way for better sampling techniques and how it would perform cross domains?

**Reasons To Accept:**

The annotated data collected based on this taxonomy can help with designing better counterargument frameworks to help promote safe conversations and content moderation operators. Further, the authors provided some analysis of the conversational outcomes based on the countering strategy (author, content, or non-counter). They provide a preliminary attempt at automatic category classification and counterspeech sub-type classification, with scope for potential improvements in future work.


**Reasons To Reject:**

The authors present an annotated dataset based on a proposed framework and present an analysis of this dataset. However, they do not compare their formalism with existing proposed formalisms like ([1]. [2]) and datasets like CONAN. Further, any human study of how the formalism presents itself in a controlled study could be interesting vs in the wild like Reddit.

[1] Allaway, E., Taneja, N., Leslie, S.J. and Sap, M., 2023. Towards countering essentialism through social bias reasoning. arXiv preprint arXiv:2303.16173.
[2] Maarten Sap, Saadia Gabriel, Lianhui Qin, Dan Jurafsky, Noah A Smith, and Yejin Choi. 2020. Social bias frames: Reasoning about social and power implications of language.


**Reproducibility:**

3: Could reproduce the results with some difficulty. The settings of parameters are underspecified or subjectively determined; the training/evaluation data are not widely available.

**Reviewer Confidence:**

4: Quite sure. I tried to check the important points carefully. It's unlikely, though conceivable, that I missed something that should affect my ratings.

---

> ### Author Rebuttal · Authors · 2023-08-27
>
> **Reasons to Reject**
>
> RE: They do not compare their formalism with existing proposed formalisms like ([1]. [2]) and datasets like CONAN.?
> * Thank you for the references. We found them relevant and will add them to the related work.
> * In the domain of counterspeech, CONAN and the work by Mathew et al., (2019) mainly built on or simply followed the taxonomies by Benesch et al., (2016) with slight modifications. We conducted detailed analyses of these taxonomies to ensure the new taxonomy covers all different types of counterspeech. There are overlaps in some categories (i.e., line 193, 208, 248, 258), however, the previous taxonomies fail to convey some important information. Most notably:
>     * the intensity of hostile tones does vary: intelligence insults [Insult] are more uncivil than asking someone to leave the discussion [Accusation].
>     * the strategies of positive tones can be different: kindly provide suggestions [Request] and show sympathy [Sympathy].
>
> * Our work has two salient distinctions: (1) we indicated whether the counterspeech addresses the author of hate speech or the content (i.e., in Table 2) and (2) we formalized the taxonomy based on argumentation theories (i.e., line 165-172). These yield several advantages:
>     * In counterspeech generation, Counter-content is generated by focusing on the content of hate speech, while Counter-author may have greater flexibility.
>     * In the evaluation, the relevance of the content in counterspeech to hate speech is a crucial factor for quality evaluation. But this factor may not be necessarily required for counterspeech.
>     * For implications, Counter-content might provide guidance on what and why the content is inappropriate. While Counter-author is usually not that informative.
>
> * Thank you for the suggestion, we will expand Table 2 and the discussion in Section 2 in the extra page if the paper is accepted.
>
> **Questions for Authors**
>
> RE: How do you propose to improve on these models besides naive finetuning or augmentation with EDA? Do you see a way for better sampling techniques and how it would perform cross domains?
> * These are insightful suggestions. We will explore different methods in our future work. Specifically, we are very interested in experimenting with different Parameter-Efficient Fine-Tuning methods, for example,  prompt tuning and prefix tuning. We are also very curious about the performance of the latest LLMs (Llama 2, T5, etc.) on our dataset.
> * For cross-domain performance, we will acknowledge in our limitations that replies on Reddit may not be representative of all platforms. Nevertheless, the methodology we employed in establishing the taxonomy can be extrapolated to other datasets.
>
> =====
> We refer the reader to Appendix C for the information necessary to replicate the results. We will also make our dataset and code publicly available on GitHub.

---

### Meta-Review · Area_Chair_sBgy · 2023-09-14

**Recommendation:** 4

**Metareview:**

Reviewers generally found that this was an interesting and fairly sound paper: The topic of focus is important, and the proposed counterspeech taxonomy is a useful addition that other researchers in this area could adopt/build on.

The main concern the reviewers had was with the lack of comparison to other frameworks for counterspeech. I think that addressing this concern, as the authors have started to do in their responses, is important, because it would better situate their work in this area. The meta-reviewer’s opinion is that the choice to work from an argument-mining perspective is super interesting and compelling; comparisons to other taxonomies would have the additional benefit of clarify why that’s the case. (Per one of the reviewers: how does the new taxonomy “add knowledge to what we already knew?”) I suggest that the authors heed our suggestions to expand their discussion on this point (alongside other reviewers’ comments).

---

### Decision · Program_Chairs · 2023-10-07

**Decision:**

Accept-Main

**Comment:**

Reviewers generally found that this was an interesting and fairly sound paper: The topic of focus is important, and the proposed counterspeech taxonomy is a useful addition that other researchers in this area could adopt/build on.

The main concern the reviewers had was with the lack of comparison to other frameworks for counterspeech. I think that addressing this concern, as the authors have started to do in their responses, is important, because it would better situate their work in this area. The meta-reviewer’s opinion is that the choice to work from an argument-mining perspective is super interesting and compelling; comparisons to other taxonomies would have the additional benefit of clarify why that’s the case. (Per one of the reviewers: how does the new taxonomy “add knowledge to what we already knew?”) I suggest that the authors heed our suggestions to expand their discussion on this point (alongside other reviewers’ comments).